# Research on Public Environmental Perception of Emotion, Taking Haze as an Example

**DOI:** 10.3390/ijerph182212115

**Published:** 2021-11-18

**Authors:** Qiang Bao, Xujuan Zhang, Xijuan Wu, Qiang Zhang, Jinshou Chen

**Affiliations:** 1College of Computing, Illinois Institute of Technology, Chicago, IL 60616, USA; qbao1@hawk.iit.edu; 2College of Computer Science & Artificial Intelligence, Lanzhou Institute of Technology, Lanzhou 730050, China; zhangxujuan0601@163.com; 3Department of Computer Science and Engineering, Northwest Normal University, Lanzhou 730070, China; 2020211982@nwnu.edu.cn (X.W.); zhangq@nwnu.edu.cn (Q.Z.)

**Keywords:** convolutional neural network, environmental perception of emotion, weighted networks, deep learning

## Abstract

Ecological and environmental problems have become increasingly prominent in recent years. Environmental problems represented by haze have become a topic that affects the harmonious ecology of human beings. The trend of this topic is on the rise. People’s perception of the environment after the impact of haze has also changed. A real-time grasp of the dynamic public environment perception of emotions is often an important basis for environmental management departments to effectively solve environmental problems through public opinion. This article focuses on the problem of the public perception of emotional changes, which is caused by fog and hazy weather, proposes an environmental emotion perception model, using Weibo comment data about fog and haze as environmental perception data, and analyzes the impact of fog and haze on the public in four seasonal time dimensions. The post-environment perception of emotion changes: the results show that in spring, the public’s environmental perception of emotions is mainly negative emotions at the beginning of the season; in summer, positive emotions become dominant emotions; in autumn, the public’s environmental perception of emotions is dominated by negative emotions that increase substantially; and in winter, the dominant environmental perception of emotions of the public is still negative. This theory provides support for research on social emotions and public opinion behavior.

## 1. Introduction

Ecological and environmental problems have long been a huge challenge facing mankind. Since China’s reform and opening up, the economy has developed rapidly; with rapid economic development, there are ecological and environmental problems. Under the threat of ecological and environmental problems, social unrest will occur, and human beings will continue to lose their living space of beneficial survival. Problems such as frequent outbreaks of a haze ecological environment pose a great threat to people’s physical and mental health [1]. The haze problem has now risen to a social problem, so the effective management of the haze problem has become a top priority in the management of environmental problems.

Environmental perception is an individual’s processing collection or processing of the impression information formed in his/her mind after being influenced by or acted on by his/her surrounding ecological environment, thereby forming the individual’s perception of changes in the ecological environment. The basic framework had been proposed by John R. Gold [2] and KP Burneett [3]. The environmental perception of emotion is the emotional change produced by an individual under the influence of an ecological environment. The current literature, however, seems focused on factors or characteristics that affect perception. Few studies have focused on the perception of environmental emotion in environmental perception to study the public’s perception theory after being affected by the environment. Only a detailed study of the environmental perception of emotions can further reveal the public’s environmental perception theory and the orientation of public opinion after the environmental impact. Therefore, if the ecological environment problem is effectively solved, it is beneficial to the public. Correct public environmental perception can create good public environmental behavior [4], but good public environmental behavior will inevitably be drawn by positive environmental perception of emotions. Therefore, a good environmental awareness is an important prerequisite for the public to effectively implement environmental protection policies and actively respond.

Traditional perception data acquisition methods collect public emotion perception data after being affected by the ecological environment through third-party observation methods such as interviews and questionnaires. However, this method of acquiring perceptual data has problems, such as a small volume of data, poor representation, too small of a collection range, and a strong, static cross-section. Since the method of collecting data is more structured, the data collected are more inductive to the objects collected. The perception data obtained by this method cannot correctly map the public’s most natural and true perception of emotions. Nowadays, with the rapid development of Internet technology, the online world has gradually become a more important part of humans’ real-life world. Coupled with the rapid development and widespread popularization of mobile devices, wireless networks, and various integrated perception computing technologies, these technologies and devices have gradually integrated into people’s daily life and become an inseparable part of human life. It enables people to live in a mixed environment of the Internet, sensor networks, and a wireless network. The digital footprint left by humans in a mixed network environment converges into a complex picture of individual and group behaviors, which helps to understand and support human social activities [5]. Through the scientific understanding of humans and their social behaviors, we can improve the quality of our daily lives, such as reducing traffic congestion, limiting the spread of disease, and optimizing public resource scheduling [6]. As an important social media platform, Weibo has become one of the typical representatives of the rapid development of Internet technology in the online world. Compared with other online social media, Weibo has the characteristics of real-time information update, an open information public opinion field, and the fast sharing of information. With the advent of the Web 2.0 era, Weibo, a new type of social networking platform, has developed rapidly [7]. It has become more and more influential in today’s social life, and has now become one of the most important social media platforms in China [8]. Thus, the Weibo platform has become an increasingly more mainstream public perception voice channel.

There are still problems in the acquisition of traditional perception data. At the same time, few studies currently use the environmental perception of emotion in environmental perception as the research focus with which to study the public’s perception theory after being affected by the environment. This article utilized the Weibo platform to obtain the public’s Weibo comment data after being affected by haze, and used them as perception data. Through the establishment of an environmental emotion perception model, the deep learning method was used to mine the public’s impact after hazy weather and the changing pattern of the environmental perception of emotion over time.

## 2. Methods and Related Work

This article mainly obtains Weibo comment data about haze through the Weibo platform. After the data cleaning process, the data are vectorized and the deep learning model is used to train the classification model of the data that have been marked with emotional polarity labels. After the trained model reaches a certain evaluation standard the data of unlabeled labels based on this model can be input, and the classification model can automatically calculate the label of the new data of unlabeled labels [9]. This research hypothesis can draw conclusions, draw an environmental sentiment prediction model, obtain the data after use, analyze how the weather affects social public sentiment, conduct environmental prediction sentiment analysis through the classification model, provide relevant suggestions for the environmental protection department, and grasp the dynamic public-perceived emotions in the environment, which is important for environmental management departments to effectively solve environmental problems through public opinion and emotions, as shown Figure 1.

The specific research tasks will to be done in this article are as follows:(1)Construction of an environmental emotion perception model. Based on the current research background of social perception computing and deep learning, an environmental emotion perception model of the public after being affected by fog and hazy weather is constructed.(2)Acquisition of perceptual data. Use web crawler technology to obtain Weibo comment data about haze from Weibo users from 1 January 2018 to 31 December 2018 on the Weibo platform as the data source of this study.(3)Data preprocessing. Preprocess the crawled Weibo comment data, including advertisement removal, garbled information and special symbol removal, low-frequency words and stop words removal, redundant information processing, etc.(4)Annotation of emotional polarity. It is necessary to manually label the emotion polarity of each piece of Weibo comment data in some of the data, to produce a training set used to train the model and a test set used to verify the model.(5)Extraction of label data features. We need to segment the data and then vectorize the words. Prepare for data entry of the model.(6)Construction of an environmental perception of emotion classification model. Use deep learning to construct a classification model and continuously adjust the model parameters, then compare the model with the traditional machine learning classification model so that the model can achieve the classification accuracy required by the experiment.(7)Analysis and research on the environmental perception of emotions. Using the trained model, the comment data that have not been manually annotated are annotated with emotional polarity, combined with the time series analysis of public sentiment changes in the sentiment perception of haze.

## 3. Results Analysis

As haze continue to interfere with people’s physical and mental health, the ecological crisis has gradually escalated into a perception crisis in people’s minds. Therefore, in hazy weather, the public expresses themselves very dynamically. Social media and smartphones, with social tools such as Weibo, provide a convenient method that allows for the expression of public emotions and perceptions after experiencing hazy weather. These public expressions and recordings can ultimately form environmental perception data. These environmental perception data can basically reflect the public’s environmental demands and responses to environmental protection policies, in addition to providing public opinion guidance for environmental changes. Therefore, these perception data have relatively high accuracy in reflecting public opinion.

### 3.1. Data Sources

In order to fully obtain the public’s perception data after being affected by haze, we used “haze” and “PM2.5” as search keywords and used web crawlers to crawl on Weibo from 1 January 2018 to 31 December 2018. Weibo comment data were released by Weibo users on 31 December. The work of cleaning and preprocessing the data mainly includes the removal of advertisements and deduplication, etc. Finally, 202,795 effective data are obtained. The pre-processed data is divided into spring (March–May), summer (June–August), autumn (September–November), and winter (December–February), according to the method of climatological division of the four seasons. According to the time dimension of the seasons, the changing pattern of the public’s environmental perception after being affected by haze is explored. Through simple statistics, the amount of environmental perception data of the relevant haze in the four seasons is shown in Figure 2.

It can be seen from Figure 2 that the public is affected by seasonal changes in haze, and the amount of perception data corresponding to the related haze also changes accordingly. From spring to summer, the number of perceptions decreased significantly, and the magnitude of the decline was large. From summer to autumn, the number of perceptions began to rise rapidly and significantly. From autumn to winter, the number of perceptions increased relatively slowly, but the highest number of perceptions was in winter. The amount of perceived data represents the magnitude of the public’s perception of their surrounding environment after being affected by haze. Since the occurrence of haze is obviously affected by seasonal changes, summer is the low season of haze, and the public’s environmental perception has a strong downward trend, while winter is generally the season of high haze occurrence, and the public’s environmental perception is stronger.

### 3.2. Data Analysis

It is impossible to grasp the public’s environmental emotion perception and the orientation of public opinion after being affected by haze, assuming that the intensity of the public’s environmental perception is considered alone. We need to analyze the public’s environmental perception of emotions and environmental perception data from a more granular research perspective. Only though further studies on the polarity of the perceived emotions can we accurately reveal the trends and changes of public sentiment and sentiment in the public environment.

#### 3.2.1. Classification Analysis of the Environmental Perception of Emotion Polarity

In 2001, Huettner et al. [10] analyzed the emotional color of words after tagging sentiment dictionaries for sentiment analysis. Hatzivassiloglou and Mckeown [11] had classified sentiment tendencies on English vocabulary. Sentiment analysis is the analysis and calculation of the sentiment opinion and sentiment attitude of an entity in the text [12]. The literature [13] believes that the main purpose of sentiment analysis is to clarify the reviewer’s attitude response to the object of the review, and its basic task is to give positive, neutral, and negative responses to the reviewers for their attitude information at a certain level. These three attitudes responded. Therefore, this article also uses these three emotion polarities as environmental perception of emotion classification labels. After the original dataset is shuffled in order, 20% of the data is randomly selected for each season for artificial emotional polarity labeling. In order to ensure the accuracy and effectiveness of emotional polarity labeling, a special emotional polarity labeling team has been formed. The theoretical method of data labeling was completed under the guidance of expert groups in the field of the environment and psychology. Data labeling was completed by six master students in psychology, two master students in environmental science, two doctoral students in environmental science, and four doctoral students in psychology. The marked data labels were collated and compounded, and finally a total of 40,570 pieces of manually marked emotional polarity data were obtained. According to preliminary statistics, the distribution of the three emotion polarities in each season is shown in Figure 3 below.

Figure 3 shows that in the four seasons of emotional polarity, positive emotions and the other two emotions displays a negative correlation with each other. When the intensity of positive emotions begins to increase, the intensity of both negative emotions and neutral emotions weakens. Additionally, the volume of data of negative emotion polarity in other seasons except summer is obviously higher than that of the other two emotions. The total amount of the three emotions manually marked in the four seasons is consistent with the change in the total amount of the original perception data in the four seasons, which basically conforms to the principle of random sampling in the total sample. We can further calculate the proportion of the three emotional polarities in each season, as shown in Figure 4.

Figure 4 shows the proportion of emotional polarity in each season. Negative emotions account for the highest proportion in the spring, followed by neutral emotions, and positive emotions account for the lowest percentage. However, in the summer, the proportion of positive emotions increases rapidly and reaches the highest level. Both neutral and negative emotions are declining, but the decline of negative emotions is larger than that of neutral emotions, and negative emotions are minimized. From summer to autumn and then to winter positive emotions begin to decline sharply; the decline from autumn tends to be flat and they drop to their minimum in winter. During this process, negative emotions are on the rise and rise to their highest point in winter; only the magnitude of the change in the level of neutral emotions is not obvious. It can be seen from the whole figure of the proportion of emotional polarity that after the public is affected by haze, positive emotions are negatively correlated with negative emotions, while neutral emotions tend to change steadily.

#### 3.2.2. Classification Analysis of Environmental

Based on the total ratio of the three kinds of environmental perception of emotion polarity in different seasons, in order to further understand the more fine-grained environmental perception of emotion polarity change trend and public opinion trend in all the perception data after the public is affected by the haze, all environment-aware data are used for quantitative research on perceived emotions.

In the traditional sentiment classification research, dictionary-based sentiment classification methods rely too much on sentiment dictionary construction and rule-making quality, and most machine learning sentiment classification methods are susceptible to the effects of manual annotation quality [14] on their classification models. The application effect applied in different scenes cannot be guaranteed [15]. Since deep learning is different than most machine learning sentiment classification methods, deep learning can more comprehensively extract very complex mapping functions with deep root semantic levels.

Convolutional neural networks (CNNs) are one of the most representative types of artificial neural networks (ANN) in current deep learning. They are similar to the receptive field mechanism in biology. It is a feedforward neural network [16], mainly composed of input layers, convolutional layers, pooling layers, and fully connected layers [17], as shown in Figure 5.

The concept of a CNN was first proposed by Fukishima [18]. A CNN has a more obvious classification effect in solving the sentiment classification problem of sentence types [19], and in this document, Kim proposed for the first time and realized the purpose of text classification based on a CNN classification model, and showed that the classification effect of the classification model has a higher accuracy rate than the best classification method at the time. In the literature [20], the accuracy of using a CNN for short text feature extraction and sentiment classification is 5% higher than that of SVM and RNN models. The literature [21] applied a CNN to Chinese Weibo sentiment classification and found that its accuracy rate was improved by 2.4% compared with SVM. Therefore, the excellent performance of CNNs in text classification [22] and sentiment classification [23] has been favored by various experts and scholars in the professional field [24]. Therefore, this paper considers using a CNN as a classification model for experiments.

When using the CNN model, the first step is to perform feature extraction on the training data, that is, label embedding neural network language model training for word embedding. Weibo comment data are short pieces of text with a brief content, generated through a social networking platform. The text as a whole has a large number, scattered semantic features, much spoken vocabulary, and complicated new words for the network. Word2vec is a word vector training tool that visually expresses semantic relationships with a numerical distribution [25,26], which solves problems such as vector dimension disasters and semantic gaps between words that cannot be solved by vector space and other models. The model includes two framework models, CBOW and Skip-gram. These two framework models also include two methods, negative sampling and hierarchical softmax, which can effectively reduce complexity [27]. The principle of CBOW is to use a given context to predict the occurrence probability of the current word, while Skip-gram is just the opposite. It predicts the occurrence probability of contextual words given the current word. This paper uses the Skip-gram model in word2vec to complete the word vector training for the phrase.

Another parameter that needs to be set is the dimension value k of word embedding [28]. Increasing k will make the mapping of each word in the text in the semantic space more accurate, but too high of a k value will cause the training process to overfit. Moreover, the requirements for training costs (such as hardware costs) will also increase. After comprehensive consideration, this article finally decided to set the word embedding dimension, k, to 300 (k = 300). After preprocessing and segmenting the labeled perception data, it is found that the longest piece of comment data has s = 121 words, that is, 121 is the longest text present. Word embedding was used on the remaining shorter texts and filled with zero padding to ensure that the word embedding dimensions corresponding to all words are consistent.

After the final processing, the total number of words in each comment in the blog comment data is, at most, s, and after training the phrase the k-dimensional vector of the i-th word can be expressed as {$x_{1},{\;x}_{2},{\;x}_{3},\ldots ,x_{k}$},$x_{j}$ dimension j, 1 < j < k. Finally, we vertically stack the word embedding of each piece of Weibo perception data in the form of a two-dimensional feature matrix as the input data, X, of the input layer of the CNN model. The dimension of the input word vector matrix is s×k. Then, we use the convolution kernel W ($W\in R^{h*k}$) to perform a convolution operation on the sub-matrix of the input word vector matrix X, with a size of h × k moving from top to bottom on X; h is the window of the convolution kernel convolution size. In this way, the word vector expressed in the sentence will be subjected to s h + 1 convolution operations with any convolution kernel, W, and the final convolution result is shown in Equation (1):(1)$$R_{i}=W\cdot \alpha_{i:i+h-1}$$

In Equation (1), the i-th eigenvalue after convolution is $r_{i}$, and 1 < I < s-h + 1. $a_{i:i+h-1}$ is a matrix block composed of $a_{i},{\;a}_{i+1},\cdots ,a_{i:i+h-1}$.

Generally, in the design of convolutional neural network models for image recognition tasks, one convolution layer corresponds to one convolution kernel. However, when constructing a model for text classification processing, the characteristics of the sentence exist in specific different parts due to the different length of each Weibo comment. In order to extract the complete sentence features more comprehensively, we used a convolutional layer when constructing the model. Convolution kernels of different sizes are used to obtain the local features of different sentences in Weibo comments as well as the long-range associations between the features [29]. Adding multiple convolution kernels of different sizes can extract local abstract features in different positions from different granularities more comprehensively, which can further reduce the chance of feature extraction and the parameters of the model, and enhance the generalization performance of the model. According to the needs of this experimental scenario, the experiment uses different window sizes, h, that is, three convolution kernels of different sizes are constructed empirically in the convolution layer, and the window sizes of the convolution kernel words are 2, 3, and 4, respectively, so three convolutional features are available.

After s-h+1 convolutions, each convolution operation also needs to normalize its result. We use activation functions to achieve nonlinear operations. The commonly used activation function is called a linear correction unit function, which is Equation (2):(2)$$d_{i}=f\left(r_{i}+b\right)$$

In Equation (2), f is the activation function and b is the offset, which is used to continuously optimize and adjust the weight matrix. The activation function of this paper uses the ReLu function. It can effectively speed up the training convergence speed. See Equation (3) for the definition of the ReLu function:(3)$$f\left(x\right)=\max\left(0,x\right)$$

Through the convolution operation, we have extracted the basic features of the sentence, but these extracted features are more prone to overfitting. Therefore, it is necessary to further perform a pooling sampling operation on the features extracted by the convolution, which is to convert the high-dimensional features extracted by the convolution operation into low-dimensional features. Common sampling methods include average and maximum sampling for max pooling. The sampling method used in the literature [30] is maximum sampling. Considering the application scenarios in this paper, this experiment also uses maximum sampling to extract the convolution maximum value in the result, the eigenvalue result, d’, obtained by pooling is shown in Equation (4):(4)$$d^{\prime }=\sum_{i=1}^{s-h+1}\max\left(d_{i}\right)$$

After the maximum value sampling, the parameters can be further reduced, the dependence of the model on the parameters can be reduced, and the promotion performance of the mode can be enhanced.

Finally, we will use the results of the pooled sampling to perform a fully connected operation with a dropout algorithm strategy. The dropout algorithm can make part of the feature information randomly selected and discarded in the pooled results, that is, the parameters will be randomly selected and discarded every time they are updated, which can effectively prevent overfitting during model training, thus improve the accuracy of the model [31]. In this experiment, in order to prevent overfitting during model training, half of the parameters are randomly selected and discarded each time the parameter is updated, and the corresponding value can be replaced with 0. The experiment chooses dropout = 0.5.

Finally, the fully connected output is connected to the softmax classifier to complete the multi-classification [32] task. The softmax function is also called the normalized exponential function [33], and the formula is defined in Equation (5). Its main purpose is to convert any real number vector, x, of J dimension into a real number vector with J digits and each dimension element between (0,1), where I = 1, 2, 3, …, J:(5)$$f_{i}\left(x\right)=\frac{e^{x_{i}}}{\sum_{j=1}^{J}e^{x_{i}}}$$

Since this experiment needs to classify the environmental perception of emotion into three categories, the node value output by the classifier needs to be set to 3. Finally, the 3D vector output from the convolutional neural network is normalized, and the probability of the perceived emotion category corresponding to each piece of perception data is calculated. The category with the highest probability is the emotion perception category of the piece of perception data. This completes the emotion category classification of the perception data.

In the process of convolutional neural network model training, the parameter information in the CNN model and the parameter information of the word vector in the input layer need to be trained. If N represents CNN convolution operation parameter information, V represents word vector parameter information, and W represents classifier parameter information, when $\theta =\left\{N,V,W\right\}$, the training sample set ε can be expressed as $\varepsilon =\left(c_{1},e_{1}\right),\left(c_{2},e_{2}\right),\left(c_{3},e_{3}\right),\ldots ,\left(c_{m},e_{m}\right)\}$. Then, the experiment needs to optimize the final target parameter, L, which is represented by Equation (6):(6)$$L=\sum_{i}^{m}{\mathrm{logP}(e}_{i}{|c}_{i},\theta )+\sum_{i=1}^{m}\frac{\lambda }{2k}\theta^{2}$$

In Equation (6), m is the number of perception data in the training set, $c_{i}$ is the i-th perception data to be classified, $e_{i}$ is the perception emotion category corresponding to the i-th perception data, and $P(e_{i}|c_{i}$) represents the probability that the perceptual data, $c_{i}$, is judged as $e_{i}$, emotion, by the model when the parameter θ is known.

In this experiment, the L2 regularization method is used to constrain the loss function, and the coefficient is set to 0.0001. The mini-batch gradient descent algorithm is used to prevent the training process from converging too slowly when using the traditional gradient descent method, and can avoid discarding global information. Only locally convergent solutions and other problems are obtained at the final convergence. Using the mini-batch gradient descent method will only allow a small part of the samples to participate in iterative training during each training iteration, so that it can meet the accelerated convergence and ensure that the optimal solution is found. The process of parameter update is shown in Equation (7). Among them, η is the parameter update rate, that is, the learning rate. In this experiment, choose η = 0.001:(7)$$\theta =\theta +\eta \frac{\partial L}{\partial \theta }$$

For classification problems, we often use some evaluation criteria to analyze the good performance of a model. Accuracy is the model’s ability to classify decisions correctly, and effectiveness is the model’s ability to classify decisions quickly [34]. In the binary classification problem, there are generally two error situations: predicting positive samples as negative samples, and predicting negative samples as positive samples. We list them as a confusion matrix, as shown in Table 1.

For the evaluation of the performance of the binary classification model, indicators such as accuracy rate, recall rate, and the F1 value are generally used to analyze the performance of the model. The accuracy rate is the ratio of the number of positive samples predicted as positive samples to the number of all positive samples in the total test sample. The recall rate is the ratio of the number of positive samples predicted to the number of positive samples and the number of all positive samples. The F1 value is a comprehensive consideration of the two. See Equations (8)–(10) for calculation formulas:(8)$$P=\frac{\mathrm{TP}}{\mathrm{TP}+\mathrm{FP}}$$
(9)$$R=\frac{\mathrm{TP}}{\mathrm{TP}+\mathrm{FN}}$$
(10)$$F1=2\frac{\mathrm{PR}}{P+R}$$

However, for multi-classification problems, the total number of positive samples will be much smaller than the total number of non-positive samples, so the use of a binary classification model evaluation method will make the model performance evaluation analysis incomplete and cause evaluation errors. In order to take into account all the classification categories in the multi-classification problem and make an accurate performance evaluation of the model, we generally use macro-averaging and micro-averaging [35] as evaluation indicators. The macro-average is the arithmetic average of the accuracy rate and recall rate of each classification category to find its average. See Equations (11)–(13) for the calculation formula. The micro-average is found by summing up the data of all categories and then calculating the corresponding index. See Equations (14)–(16) for calculation equations:(11)$$\mathrm{MacroP}=\frac{1}{n}\sum_{i=1}^{n}P_{i}$$
(12)$$\mathrm{MacroP}=\frac{1}{n}\sum_{i=1}^{n}R_{i}$$
(13)$$\mathrm{MacroF}=\frac{2*\mathrm{MacroP}*\mathrm{MacroR}}{\mathrm{MacroP}+\mathrm{MacroR}}$$
(14)$$\mathrm{MicroP}=\frac{\bar{\mathrm{TP}}}{\bar{\mathrm{TP}}+\bar{\mathrm{FP}}}$$
(15)$$\mathrm{MicroR}=\frac{\bar{\mathrm{TP}}}{\bar{\mathrm{TP}}+\bar{\mathrm{FN}}}$$
(16)$$\mathrm{MicroF}1=\frac{2*\mathrm{MicroP}*\mathrm{MicroR}}{\mathrm{MicroP}+\mathrm{MicroR}}$$

This indicator can evaluate the accuracy of each classification category based on the classification performance of all datasets. This experiment divides the perceived emotion into three categories, so we choose *n* = 3.

In this experiment, all the data marked with emotion tags will be distributed according to the ratio of 5:1 between the training set and the test set [36], using 5/6 of the data to support the establishment of the model [37]; the remaining 1/6 is used to test the model’s performance [38]. The experimental results were tested by six-fold cross-validation to test the performance of the classification model. The advantage of cross-validation is that the randomly generated sub-samples are repeatedly trained and verified, the results being verified once each time, and the average value is finally taken as the result. The six parameters of MacroP, MacroR, MacroF, MicroP, MicroR, and MicroF are shown in Table 2.

In order to further verify the performance of the model used in this experiment, we compare the classification effect of this experimental model with the SVM model with a better classification effect in traditional machine learning. The comparison results are shown in Table 3.

By comparing the six parameters, it is shown that, compared with the mainstream SVM model used in machine learning, the CNN models with different convolution kernels used in this paper have significantly improved the classification performance of the environmental perception of emotions in the microblog comment Weibo data obtained in 2018.

#### 3.2.3. Application Analysis of Environmental Perception of Emotion Model

In order to further apply the model, after the above experimental analysis, this paper finally uses a variety of trained convolution kernel convolution neural networks to perform the calculation and labeling of perceived emotion tags on the remaining large amount of unlabeled emotional tag blog data. Finally, we acquire the Weibo comments about haze in the whole year of 2018. The data volume change trend of the three kinds of environmental perception emotions in each day is shown in Figure 6, where the vertical axis represents the Weibo comments about haze quantity. The abscissa axis represents time.

Figure 6 reflects the trend of the public’s perception of mood each day after being affected by hazy weather. It can be seen that, in different seasons, there are differences in the trend of the environmental perception of emotions of the public after being affected by haze. Among them, the amount of blog data for a certain emotion on a certain day represents the perceived intensity of the environment’s perceived emotion after the public is affected by haze on that day. In addition, on the basis of the known environmental perception emotion trend, weakening the noise of the perception emotion trend makes the public environment perception emotion trend more obvious and relatively smooth. Divide the time according to the season and obtain the five-day moving average chart, as shown in Figure 7, Figure 8, Figure 9 and Figure 10.

It can be clearly seen from Figure 7, Figure 8, Figure 9 and Figure 10 that under the influence of haze, the public’s emotional responses differ greatly in the four seasons.

Among them, Figure 7 shows the public’s emotional perception of the environment in the spring. The relative proportion of negative emotions of the public is the highest throughout the spring, but the perceived intensity of negative emotions tends to gradually weaken from April, while the perception of neutral emotions is similar to that of negative emotions and gradually weakens from April as well. The overall perceived intensity of positive emotions has a tendency to increase slowly. Before May, positive emotions have been less prevalent than neutral emotions. It has only been since May that the perceived intensity of positive emotions exceeded the other two emotions, indicating that the dominant perceived emotion by the public in the spring gradually changed from negative emotion to positive emotion.

Figure 8 shows that the changes in the public perception of environmental emotions during the summer. Obviously, the public’s positive emotions account for a relatively high proportion, but the positive emotion perception has a slowly decreasing trend throughout the summer. From 7 August, positive emotion perception is lower than that of neutral emotion, while neutral emotion perception is relatively stable. The change is not obvious. Although the proportion of negative emotions is low, there has been a gradual increase in the trend from August. After August, the changes in the three emotions show a tendency of intertwining, indicating that in the summer the public’s environmental perception is mainly dominated by positive emotions. However, due to the increase in negative emotions in the latter part of the season, neutral and positive emotions begin to weaken.

Figure 9 reflects the changes in the public’s environmental sentiment trends in autumn. Among them, the perception of negative emotions is continuously increasing, while the rate of increase is also increasing. The perception of neutral emotions is also increasing, but this increase is weaker than that of negative emotions. Positive emotions are slowly and steadily weakening. It shows that the public’s environmental perception in the autumn fluctuates greatly, and is mainly dominated by negative emotions.

Figure 10 reflects the environmental perception of the public after being affected by haze in winter. It can be seen that the overall change trend of the public’s negative emotions and neutral emotions is mainly a sharp decline, but the proportion of negative emotions is always the highest only in the days close to 10 February; the proportion of neutral emotions is close to the proportion of negative emotions. The perception intensity of positive sentiment first increased slowly, and gradually began to weaken gradually by mid-January, and its proportion was always small. Overall, the perceived intensity of positive emotions in the environmental perception of emotions in winter has never exceeded that of negative emotions and neutral emotions, indicating that the public has a stronger perception of negative emotions after being affected by haze in winter.

## 4. Conclusions

Environmental issues highlight the public’s response to environment impacts. The public’s environmental perception can reflect the public’s attention to environmental changes and public opinion guidance. Correctly grasping the public’s environmental perception of emotions and public opinion guidance is beneficial to the Environmental Protection Department, allowing for the effective promotion and implementation of environmental protection policies. It is also a form of appropriate feedback from the public on the effect and satisfaction of the policies implemented by the Environmental Protection Department.

From the perspective of perceiving big data, this paper uses environmental perception data from social networks to reveal the public’s emotional response law of environmental perception after being affected by haze from a new research angle. Experiments show that the mood of the public fluctuates significantly with seasonal changes after being affected by hazy weather. It reflects the public’s cognitive response and the direction of changes in public opinion after being affected by haze. On the whole, in the spring, the public’s environment-perceived emotions were mainly negative emotions at the beginning of the season, followed by neutral emotions. As the spring approached the end, positive emotions gradually increased became the dominant emotions. Therefore, the Environmental Protection Department should promptly guide the public’s negative environmental perception emotions to enhance the public’s environmental awareness. In summer, positive emotions become the dominant emotions. Relatively speaking, the perception of negative emotions is weak, and there is a tendency for it to slowly increase before the fall, because summer is the season of low haze and the public is satisfied with its surrounding ecological environment. The proportion of negative and neutral emotions is relatively low, and the public’s positive emotions are relatively high. When the public’s environmental perception of emotion is positive, it is easier to accept the guidance of relevant environmental protection decisions, so the Environmental Protection Department can appropriately implement environmental protection decisions at this time, so that the public can adapt to environmental protection decision guidance to the greatest extent. Negative public sentiment increased substantially in the autumn and became the dominant emotion. As the positive sentiment slowly weakened, the negative public sentiment perception in the entire autumn season was relatively strong. Near the end of the season, the intensity of the public negative sentiment reached its highest level. Therefore, when the public’s perception of environmental sentiment in the Environmental Protection Department is in a strong negative state, it is necessary to enhance the public’s appeal for a better environment, increase public awareness of environmental protection, and give appropriate public opinion guidance under certain circumstances. In the winter, negative emotions are the dominant emotions in the public’s perception of emotions, but they are similar to neutral emotions, and the downward trend is obvious. Although positive emotions change slowly, the public negative emotions have been significantly eased and gradually stabilized. Therefore, the Environmental Protection Department should establish the public’s environmental protection concepts, implement appropriate environmental protection policies, and should also ease the public’s negative environmental perception of emotions, rather than blindly panic.

## 5. Discussion

This paper used deep learning models to analyze the public’s emotional perceptions after being affected by the environment during different seasons. It is supported by current big data background and based on the perspective of social perception research. The recognition and classification of positive, neutral, and negative emotions finally revealed the public opinion guidance of the public’s environmental perception of emotions in the four seasons, and provided corresponding decision support for the Environmental Protection Department according to the corresponding public opinion guidance. However, this article only visually studied the public opinion orientation under the influence of hazy weather based on positive, neutral, and negative emotional polarity, and does not further subdivide the emotional polarity. The public’s environmental perception of emotions is different, and it is also affected by other factors, such as holidays. This article does not consider these factors. At present, there is no strict delimitation standard for the classification of the environmental perception of emotions. More authoritative and in-depth environmental perception emotion standards need to be further studied. In the follow-up research, it will be extended step by step, and a more in-depth study will be conducted on the analysis of the environmental perception of emotion laws, which will provide a new research perspective for the study of ecological environment perception.

## Figures and Tables

**Figure 1 ijerph-18-12115-f001:**
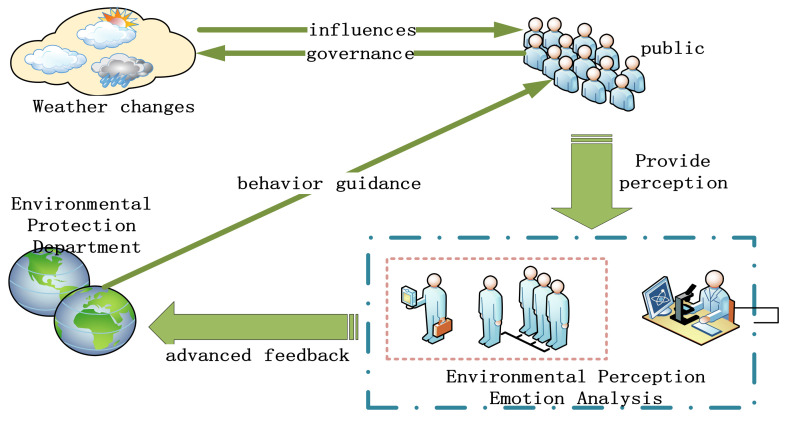
Environmental emotion perception model.

**Figure 2 ijerph-18-12115-f002:**
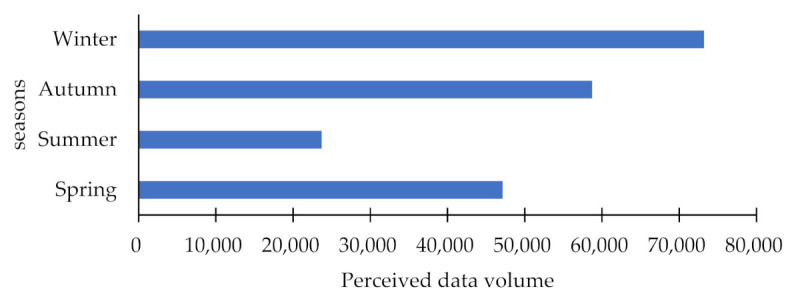
Amount of perceived data in the four seasons.

**Figure 3 ijerph-18-12115-f003:**
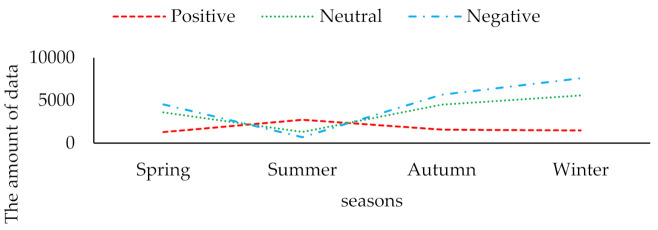
The amount of data marked by artificial emotion polarity.

**Figure 4 ijerph-18-12115-f004:**
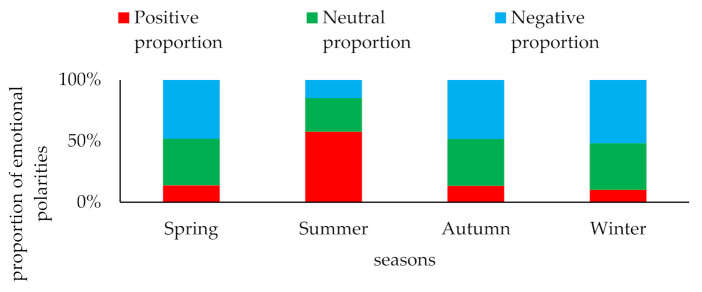
Proportion of emotional polarities by season (%).

**Figure 5 ijerph-18-12115-f005:**
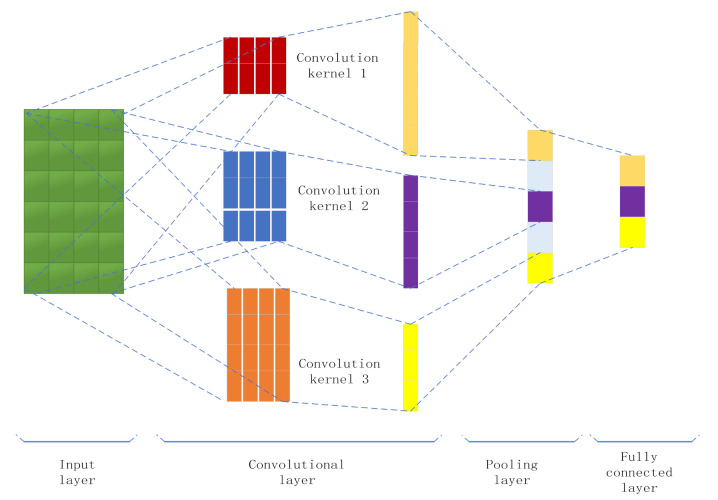
CNN structure diagram of multiple convolution kernels.

**Figure 6 ijerph-18-12115-f006:**
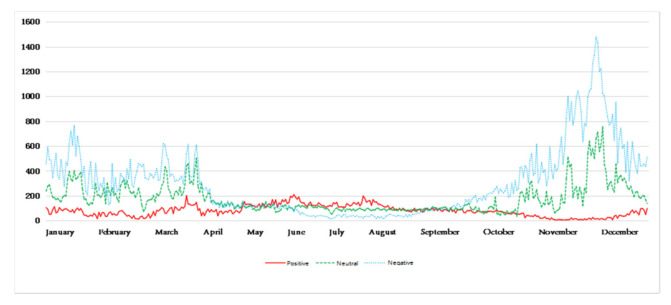
Change graph of environmental perception.

**Figure 7 ijerph-18-12115-f007:**
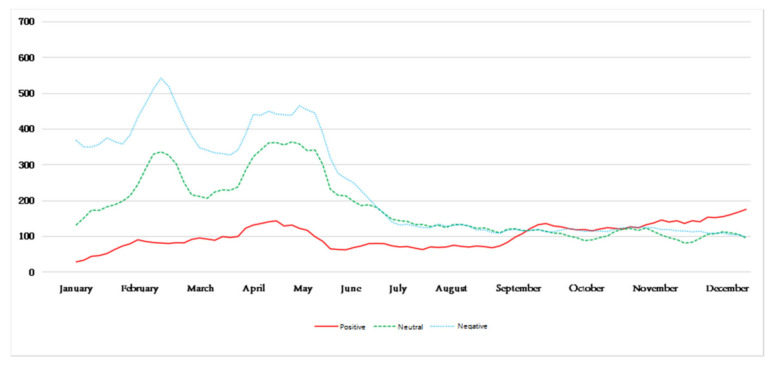
Five-day moving average of perceived emotions in spring.

**Figure 8 ijerph-18-12115-f008:**
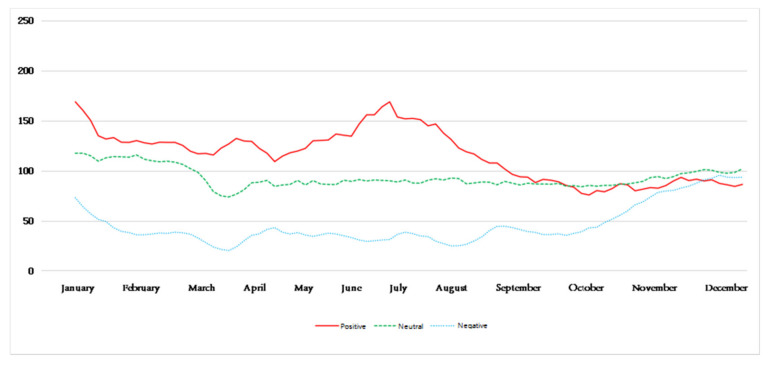
Five-day moving average of perceived emotions in summer.

**Figure 9 ijerph-18-12115-f009:**
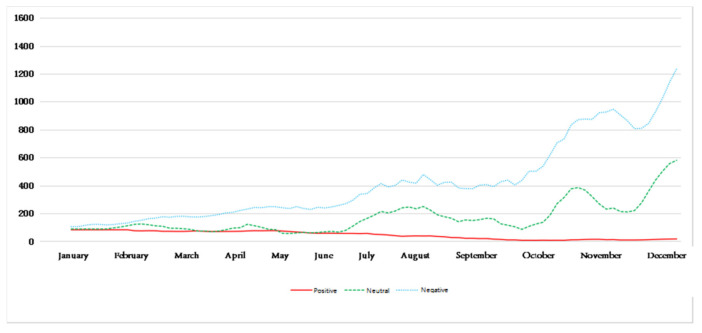
Five-day moving average of perceived emotions in autumn.

**Figure 10 ijerph-18-12115-f010:**
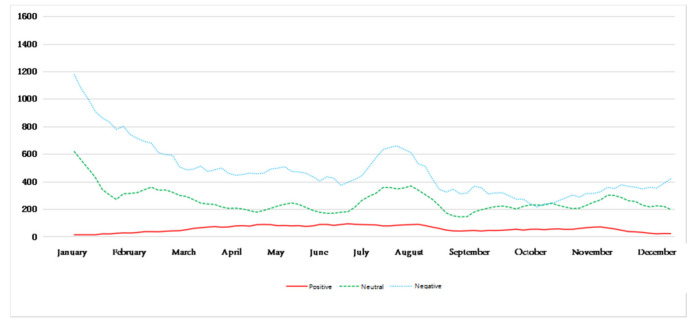
Five-day moving average of perceived emotions in winter.

**Table 1 ijerph-18-12115-t001:** Confusion matrix list.

	Positive Example	Counterexample
Positive example	Treat positive examples as positive examples (TP)	Treat positive examples as negative examples (FP)
Counterexample	Treat negative examples as positive examples (FN)	Treat counterexamples as counterexamples (TN)
Total	TPR = TP/(TP+FN)	FPR = FP/(FP+TN)

**Table 2 ijerph-18-12115-t002:** Cross-validation result analysis table.

id	MacroP	MacroR	MacroF	MicroP	MicroR	MicroF
1	88.67%	77.27%	84.96%	88.79%	79.11%	88.79%
2	89.01%	76.98%	85.67%	89.25%	78.54%	88.42%
3	88.86%	76.56%	85.55%	88.71%	78.37%	89.12%
4	88.75%	77.23%	85.91%	89.37%	79.18%	88.37%
5	88.93%	76.77%	84.63%	88.93%	78.75%	88.63%
6	88.78%	76.45%	84.71%	89.55%	79.27%	88.49%
Average	88.84%	76.88%	85.24%	89.10%	78.87%	88.64%

**Table 3 ijerph-18-12115-t003:** Model comparison results.

	MacroP	MacroR	MacroF	MicroP	MicroR	MicroF
SVM	81.76%	72.67%	80.56%	83.85%	75.88%	81.77%
CNN	88.84%	76.88%	85.24%	89.10%	78.87%	88.32%

## Data Availability

Data sharing not applicable.

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
