# Peer review of "Research on Public Environmental Perception of Emotion, Taking Haze as an Example"

_ijerph, 2021, doi:10.3390/ijerph182212115_

Round 1

Reviewer 1 Report

This paper puts forward a viewpoint of considerable research value, especially in the discussion of public’s emotional perception and the construction of the emotional perception model. However, on the premise of constructing this concept, the collection of Weibo comment information may need to consider its scientificity and representativeness of data acquisition. A few suggestions are show as below;

1. Weibo has existed for a long time. Therefore, it is better for this paper to explain the development and functions of Weibo to help readers understand this social application. In addition, the reasons why Weibo has advantages over other social software should be explained too.

2. Whether the amount of data collected from a small amount of information is sufficient or not and accurate enough in real time or over a period of time, so as to put forward a relatively correct information of public’s emotional perception. Also, to employ another way to cross compare or check the data obtained at the same time may be helpful in accruing more accurate result of public’s emotional perception for further application.

3. The feedback and comment information of Weibo varies from individual to individual. Therefore, how to judge the correct or available information among massive information to establish the public's emotional perception caused by climate change at that time is important. And how to measure the amount of data been processed that are valuable and with high representativeness for the public are curial too. Besides, it is better to develop basic data standard quantity, so that a more representative data analysis can be carried out under the condition of obtaining effective data and meeting this measurement standard for data analysis.

4. Please explain the proportion of students in the composition of the emotional polarity labeling team and the functions and work assignments of all members, so as to make the composition of the team scientific and reasonable.

Author Response

非常感谢您对本文提出的问题和建议。我们根据您提出的问题和建议对论文进行了相应的修改。详情请参阅附件。

向你致以最良好的祝愿。
所有作者。

Reviewer 2 Report

This is a very interesting article and I commend the authors for looking at a new research technique for public environmental awareness and perceptions.  

I found a number of places that seems unclear and where the formatting/grammar need to be improved.

Abstract

(13) Its popularity has also been rising - not sure what you mean by this 

(14-16) Real-time mastering dynamic public environment perception emotions ????  What do you mean by this?  A method in real-time to study dynamic perceptions of the public's emotions on environmental topics?

(16-20) First sentence is a run-on sentence and needs to be rephrased and is missing its period. (20-24) also run-on sentence. End awkwardly with perception emotions Dominate emotions.  

(24-25)  This law....  I don't believe that you have discovered a law.  I believe you are proposing a theory.

Introduction

(33-35)  Sentence is vague and unclear.  What would be more serious than survival?

(40-41) consider making this more inclusive by changing to his/her

(46) Current literature, however, seems focused on ....

(50) environmental perception law - what is this?  

(51-52) unclear sentence (52) Remove Because and start with Correct

(55-56) unclear sentence and I doubt that this is a guarantee

Most of paper needs to be rewritten in past tense

(57-59) Traditional perception data acquisition methods collect public .... 

(59-62) run on sentence

(62-64) awkward The perception data obtained through these collection methods cannot ... the public's natural and ...  

(64-69) awkward and run-on sentence also doesn't have an ending.  it makes people's lives in the Internet, sensor networks and other wireless mixed network living environments..... what does it make????

(80-82)  Had period after [7] It has become.... of the most important social media platforms in China.  Thus the Weibo...

(84-87) awkward and run-on sentence

(87-91) change to past tense. This article utilized ...  method was used.... 

(1) what is the law of environmental perception emotion??????

(93) about haze (drop the the)

(94) After cleaning it,  --- what does this mean?

(94-97) awkward sentence and a run-on

(98-100) awkward  Figure 1 displays an environmental emotion perception model based on this research.

(100-126) nice and clear

Results Analysis

(130) What is meant by - Therefore, in fog and haze weather, the public is the most dynamic subject.

(130-134) Awkward and run-on

Social media and smartphones, with social tools such as Weibo,  provide a convenient method that allows the expression of public emotions and  perceptions after experiencing hazy weather.

(134) These public expressions and recordings can ultimately form environmental perception data.

(135-138) What evidence do you have that this would then provide an accurate public opinion.  Perhaps it provides a current snapshot into public opinion but I don't think you can state that it is accurate.  The sentence also a bit awkward.

3.1 Data Sources 

(142) no date listed for end date just 2018

147-148 What is meant by law of the public's environmental perception  (is there truly a law????)

3.2 Data Analysis

(165-167) Spit comma sentence

(170) awkward  Only through further studies on the ....  can we accurately reveal....

(186) period missing after students. The doctoral... 

(193-194) change one of the word "shows" to "displays" just to add to word diversity

209) Emotions are greater and minimized.  How does it become bigger and smaller???

(217-222) Run on sentence without a complete thought

(227) While deep learning is different, ... please explain what you mean by that

(232) input Layers ....  do you need capital letter?

(237-248) Paragraph needs to be rewritten to avoid missed punctuation and run-on sentences.  Also need to finish thoughts - favor[24].  Favor what?

(250-252) - Avoid first person language -- you first need

(267-269) Awkward sentence please rewrite

(271) longest text in the text.  perhaps change to longest text present.

(272-274) avoid first person and change to past tense.

Word embedding was used on remaining shorter texts and filled with zero padding to ensure.... 

(312)  Not sure what you mean and I think you need a period

... features extracted by convolution, that is, convolution  The high....

(326)  Thus improving the accuracy of...  

 Please change using the past tense in general for the paper.

(416-418)  The Weibo comments on haze in 2018 were obtained.

(422) the haze weather (remove the) 

(422-424)......seasons, There......   - this is a split comma sentence with incorrect punctuation.

(426-430) Awkward and run-on sentence that used trend or trends too many times.

(Figure 6-10)- what are the units for the y axis?  Is it number of comments?

(462-464)  This sentence is confusing and needs to be cleaned up.

Among them, the perception of negative emotions continues to increase while increasing, and neutral emotions also increasing, but the increase is weaker than the negative emotions. 

(469-470) but the proportion of negative emotions is always the highest, only 2  In recent

This needs to be cleaned up.

(470) In recent days on the 10th, the....  10th of what??

(477-480) Split comma sentence  This paper used deep learning models to analyze the public's emotional perceptions after being affected by the environment during different seasons.  It is supported by current big data.....

(484-489) Run on sentence with missing punctuation.  Needs to be rewritten starting from  "However, this article only based on the .....  factors into account."

(516) It belongs to the summer season with low haze,  (what is it?)

(523-524)  The negative public sentiment increased substantially in the autumn and became the dominant emotion.

(537) I don't think that the paper ends well with "don't blindly panic."

(599) Citation 27 -is this a complete citation? 

Author Response

Thank you very much for your questions and suggestions on this paper. We have made corresponding amendments in the paper according to your questions and suggestions. Please refer to the attachment for details.

Best wishes to you.

All authors.
